# Plant Nutrition: An Effective Way to Alleviate Abiotic Stress in Agricultural Crops

**DOI:** 10.3390/ijms23158519

**Published:** 2022-07-31

**Authors:** Venugopalan Visha Kumari, Purabi Banerjee, Vivek Chandra Verma, Suvana Sukumaran, Malamal Alickal Sarath Chandran, Kodigal A. Gopinath, Govindarajan Venkatesh, Sushil Kumar Yadav, Vinod Kumar Singh, Neeraj Kumar Awasthi

**Affiliations:** 1ICAR-Central Research Institute for Dryland Agriculture, Hyderabad 500059, India; v.visha@icar.gov.in (V.V.K.); suvana.sukumaran@icar.gov.in (S.S.); ma.sarath@icar.gov.in (M.A.S.C.); g.venkatesh1@icar.gov.in (G.V.); sk.yadav2@icar.gov.in (S.K.Y.); 2Department of Agronomy, Faculty of Agriculture, Bidhan Chandra Krishi Vishwavidyala, Mohanpur 741251, India; itsmepurabi1@gmail.com; 3Department of Biochemistry, College of Basic Science and Humanities, G. B. Pant University of Agriculture & Technology, Pantnagar 263145, India; vivekverma95@gmail.com; 4Sirius Mineral India Private Limited, New Delhi 110092, India; neeraj.awasthi@angloamerican.com

**Keywords:** macronutrients, mechanisms, micronutrients, beneficial nutrients, abiotic stress

## Abstract

By the year 2050, the world’s population is predicted to have grown to around 9–10 billion people. The food demand in many countries continues to increase with population growth. Various abiotic stresses such as temperature, soil salinity and moisture all have an impact on plant growth and development at all levels of plant growth, including the overall plant, tissue cell, and even sub-cellular level. These abiotic stresses directly harm plants by causing protein denaturation and aggregation as well as increased fluidity of membrane lipids. In addition to direct effects, indirect damage also includes protein synthesis inhibition, protein breakdown, and membranous loss in chloroplasts and mitochondria. Abiotic stress during the reproductive stage results in flower drop, pollen sterility, pollen tube deformation, ovule abortion, and reduced yield. Plant nutrition is one of the most effective ways of reducing abiotic stress in agricultural crops. In this paper, we have discussed the effectiveness of different nutrients for alleviating abiotic stress. The roles of primary nutrients (nitrogen, phosphorous and potassium), secondary nutrients (calcium, magnesium and sulphur), micronutrients (zinc, boron, iron and copper), and beneficial nutrients (cobalt, selenium and silicon) in alleviating abiotic stress in crop plants are discussed.

## 1. Introduction

In this modern era of civilization, the changing climatic scenario has altered the natural balance of the global environment [1]. The food demand over the world is increasing due to the rapid increase in the population. At the same time, the direct and indirect effects of climate change are causing several abiotic stresses to crop growth and the environment. Abiotic stresses, for instance, drought, temperature variations, soil salinity, soil alkalinity and heavy metal stresses can have overwhelming impacts on the growth and productivity of crops under different agricultural ecosystems, which may develop constraints to food security worldwide [2,3,4].

Developing solutions to deal with the increasing frequency of extreme weather events is a challenge for agricultural researchers. In its latest report (2021), the intergovernmental panel on climate change has mentioned that human activity has warmed the atmosphere, ocean and land without a doubt. They have also warned that unless efforts are in place to reduce emissions of carbon dioxide and other greenhouse gases (GHGs), global warming of 1.5 °C and 2 °C will be exceeded during the 21st century [5]. The world’s leaders committed to a number of agreed-upon aspects when they signed the historic “Glasgow Climate Pact” in November 2021. Some of these decisions include greater efforts to strengthen climate change resilience, reduce greenhouse gas emissions, and provide the necessary funding for both [6]. We should practically double crop production to meet the growing food and nutritional demands of the growing population. Though these abiotic stresses have an impact on crop growth and productivity, the nature of the crop and the stage at which the crop experiences the stress determines the amount of crop loss it will incur. Researchers around the world have been pushed to develop a long-term solution and strategy to these challenges, which are likely to increase in the future.

Poor crop yield is very common under a wide range of environmental constraints and lack of mineral nutrient availability, Nutrient imbalances in plants have a significant impact on their performance, such as growth pattern, antioxidant defence mechanisms, and tolerance to biotic and abiotic stresses [7].

Abiotic stresses share some characteristics in terms of their effects on plants and how they are perceived by the plants (Table 1). All of these abiotic factors, for example, cause osmotic stress in plant cells. Extreme temperature variations (10–15 °C) above or below an ideal environment can cause heat or chilling/freezing stress. Stress has a substantial impact on plant functions such as seed germination, growth, development, photosynthesis, and reproduction, with serious consequences for plant growth and yield. Heat/temperature stress impacts the plant’s source–sink relationship, hormones produced in the roots, and the concentration of nutrients in the plant [8]. Heat stress in plants reduces enzymatic activity, chlorophyll content, photosynthesis, stomatal conductance, transpiration rate, antioxidants and membrane stability index while increasing reactive oxygen species (ROS) production [9] (Figure 1). Chilling stress in plants affects tissue water content, membrane fluidity and chlorophyll [10]. Many plant nutrients are reported to alleviate heat stress. For example, boron is reported as an important micro-nutrient that substantially improves the activity of the antioxidant system and alleviates the toxic effects of ROS produced by heat stress [11]. Likewise, selenium (Se) is known for its structural role in glutathione peroxidase (GPX) synthesis and, therefore, protects plants from the negative effects of ROS [12]. Being an important micronutrient, Zn substantially maintains membrane permeability, and optimum Zn supply protects plants from the devastating impacts of heat stress [13].

Drought is one of the major detrimental abiotic stresses in agricultural systems. Drought imposes a water deficit, leading to scarcity in moisture availability and restricted growth and yield in crops. Water deficit stress restricts stomatal opening, accelerates the photoreduction of oxygen (O_2_) in the chloroplast and increases photorespiration, eventually leading to oxidative damage due to the accumulation of ROS in plants [21]. In response to drought stress, plants alter gene expression, produce the phytohormone abscisic acid (ABA), close their stomata, and maintain their osmotic balance [25]. On the other hand, excessive accumulation of water in soil due to heavy precipitation over a period of time, poor drainage, etc., causes soil flooding or waterlogging. Waterlogging covers plant roots, characteristically bringing about low light, hypoxia, anoxia, impaired gaseous exchange, a rise in lipid peroxidation and ROS accumulation. It also leaches out substantial amounts of essential nutrients from the soil, accumulates salts and increases the availability of heavy metals owing to the change in soil pH, creating an adverse growing situation for the plants [26].

Salt stress affects plants by restricting water uptake, disrupting biological membranes, ionic imbalance, oxidative damage and nutritional imbalance, reducing cell division and expansion, rate of photosynthesis, lipid metabolism and impairing yield traits. Apart from that, soil salinity has also hampered agricultural production. Drought-like circumstances occur when salt-stressed plants stop absorbing water. As a result, salinity lowers stomatal conductance and disturbs photosystem (PS) and photosynthetic enzymes in plants, resulting in the generation of ROS [27]. Salinity affects approximately one-fifth of irrigated lands [28]. The favourable role of some nutrients, such as silicon (Si), in improving crop salt tolerance was documented by Tahir et al. [29].

In stressful conditions, nutrient management options can be used to provide a good crop yield. Plant nutrients help to activate various plant mechanisms to alleviate abiotic stress, including activation of stress-related genes, biosynthesis of antioxidant enzymes, osmoprotectants, heat shock proteins (HSPs), detoxification of ROS, functional or structural protection of proteins, DNA repair, membrane stability, increased photosynthetic activity, decreased uptake of heavy metals, etc. Nutrient management is a viable technique for reducing stress in the environment and increasing agricultural productivity.

## 2. Nutrient Management Approaches to Alleviate the Abiotic Stresses

### 2.1. Primary Nutrients

The notable roles of primary nutrients, i.e., nitrogen (N), phosphorus (P) and potassium (K) in activations of several plant mechanisms to alleviate abiotic stress are picturized in Figure 2. Whereas crop-wise detailed impacts of these three macronutrients under stress situations are elucidated in Table 2.

#### 2.1.1. Nitrogen

In terms of structural integrity, growth, physiology, and stress reduction, nitrogen (N) is one of the most important plant nutrients [43,44]. In almost all types of crop plants, it is linked to higher chlorophyll biosynthesis, greater photosynthetic activity, and efficient solar radiation utilisation [11,45]. The amount of stress relief was shown to be controlled by the type of nitrogen supplied, either ammonium (NH_4_^+^) or nitrate (NO_3_^−^) [46,47]. Plants fertilised with NO_3_^−^ showed a stronger tolerance to heat stress than those fertilised with NH_4_^+^, according to Zhu et al. [48] and Bendixen et al. [49]. Unused solar energy can occasionally increase photooxidative damage and lipid peroxidation in high-temperature conditions when there is an N deficiency in the plant [50,51]. Furthermore, in terms of agricultural output under moisture stress, N absorption and subsequent usage by plants are critical [31,52]. In reality, nitrogen fertiliser application in crops at a lower water potential facilitates carbon partitioning, carbohydrate build-up, cellular membrane stability, osmoregulation, and antioxidative defence mechanisms, resulting in the improved overall development and reduced leaf senescence [53,54].

The presence of N is linked to the plasticity and water extraction ability of plant roots from soil, which aids in maintaining optimum relative leaf water content and improving water use efficiency in moisture-scarce environments [30,55]. Apart from this, N helps in lowering the ROS concentrations in terms of malondialdehyde (MDA) concentration and lipid peroxidation by triggering proline accumulation and enzymatic antioxidant activities with respect to superoxide dismutase (SOD), peroxidase (POD), catalase (CAT) and ascorbate peroxidase (APX) activities [33,56,57].

The cell division and expansion mechanisms are hampered by N deficiency, especially under drought conditions, which contribute to the suppression of leaf production and development [58,59]. Furthermore, restricted N supply under stress causes faster degeneration of photosynthetic pigments (chlorophyll) and enzymes (Rubisco) concentrations, resulting in a visible slowing of plant photosynthetic activity [60]. On the other hand, a number of researchers have reported accelerated cell synthesis and expansion of plant cells and xylem tissues through increased leaf area [61,62], as well as increased photosynthetic capacity with N application in moisture deficit situations [55,63]. Aside from cellular growth roles, N has been found to play a role in protein metabolism, particularly under moisture stress [64,65]. Increased N uptake and the enzyme nitrate reductase, which is involved in nitrogen absorption, have both been linked to increased N availability in plants under water stress [66,67]. Furthermore, foliar application of N fertilisers at the reproductive stage, particularly in leguminous crops under drought conditions, significantly slows abscisic acid synthesis while accelerating cytokinin production, which promotes nodulation, cell elongation, apical dominance, shoot development, photosynthetic activity, and assimilates translocation to the sink organs [68,69]. Stress alleviation helps to improve the yield or helps in yield reduction. The yield reduction due to stress varied with the crop and its stages. In wheat, the yield penalties were 5.8% compared to 16% in control [31], whereas in rice and maize, it varied from 13 to 15% yield reduction compared to 33% with no N application [33,55].

#### 2.1.2. Phosphorous

Phosphatic fertiliser application has a strong positive correlation with plant growth and development under stress. Phosphorous enhances root architecture and proliferation in soil, even in the presence of low soil moisture, which stimulates root volume and hydraulic conductance [70,71,72,73]. This improvement in root growth facilitates efficient access to moisture and nutrients by strengthening the sensing and signalling of their availability [74,75], as well as subsequent uptake mechanisms from rhizospheric soil under abiotic stress conditions [76,77,78]. It helps in the modulation of various morphological, physiological, and biochemical processes inside the plant system, leading to stress tolerance [79,80,81,82,83]. It was found that application of P in the initial stage of the wheat crop increased root growth and establishment under water stress conditions [35]. Due to uptake of P in an adequate amount, maximum numbers of fertile tillers were produced and the spike length, number of spikelets per spike and grains per spike also increased due to better photosynthesis, energy storage, transfer, cell division as well as cell elongation, resulting in a 28% yield increase [35].

Phosphorous also helps in maintaining the cell turgidity and cell membrane stability through the acceleration of stomatal conductance and rate of net photosynthesis that helps to sustain optimum leaf water potential ensuring stress tolerance [84,85,86,87]. Apart from this, incorporation of P under various abiotic stresses has a profound impact on tissue formation, carbohydrate and lipid metabolisms, chlorophyll biosynthesis, photosynthesis, production of assimilatory compounds such as adenosine triphosphate (ATP) and reduced nicotinamide adenine dinucleotide phosphate (NADPH), enzymatic functions including ribulose-1,5-bisphosphate (RuBisCo) and ATPase activities, phytohormonal functions such as ABA and indole acetic acid (IAA), energy transfer and storage, reproductive development, ROS (e.g., hydrogen peroxide (H_2_O_2_), MDA, singlet oxygen (^1^O_2_), superoxide anions (O_2_^−^) and hydroxyl radicals (OH^•^) scavenging activity, osmolytes accumulation (proline, soluble sugars and proteins) etc. [88,89]. Drought-induced ROS accumulation has been reported to be alleviated through P application by stimulating enzymatic antioxidants such as CAT, SOD, POD, APX and monodehydroascorbate reductase (MDHAR), which inevitably elevated the capacity to withstand stressful situations [90,91]. Likewise, P application has also been reported to be involved with the upregulation of nitrogenous compounds in terms of accumulation and assimilation of NH_4_^+^ and NO_3_^−^ in water-stressed crop plants [92]. P deficiency can exacerbate the intensity of abiotic stresses [90,92]. On the other hand, abiotic stresses have also been observed to end up with P deficiency and hindered activities of ATPase and phosphatase enzymes in plants apparently under drought stress [93,94], heat stress [87,95], salinity stress [79,96], acid stress [97,98].

#### 2.1.3. Potassium

Potassium (K) is a vital macronutrient for the better growth and physiological development of crop plants under abiotic stress situations [99,100]. It is essential for many basic physiological and metabolic functions of a plant system, such as photosynthesis, stomatal regulation, photosynthates translocation, carbohydrate metabolism, maintenance of cell turgidity, enzymatic activations, etc. [93,101]. The pivotal role of K in upgrading the resilience of crops in response to the adversities of several abiotic stresses has been explored in a number of the earlier literature [102,103]. This involves multiple arrays of mechanisms, including improved root physiology, leaf surface area, stomatal control, reduced transpiration, efficient osmotic adjustment, maintenance of leaf turgor, aquaporin (channel protein) function, improved enzymatic upregulations, improved nutrient uptake and utilisation, enhanced water use efficiency, prevention of drought-induced accumulation of ROS and maintaining optimal energy status of plants [38,104,105].

Potassium prevents oxidative damage to cells by preventing ROS accumulation in terms of O_2_^−^, ^1^O_2_, H_2_O_2_, OH^•^, peroxy radicals (ROO^•^), alkoxy radicals (RO^•^), organic hydroperoxides (ROOH) and MDA [106,107] by virtue of activating a series of antioxidant enzymes such as SOD, POD, APX and CAT [108,109]. In fact, under moisture stress, K has been shown to have a favourable impact on relative leaf water content and water use efficiency [41,105,110]. Root proliferation is aided by K application in stress situations, resulting in increased water intake by plants [111]. Thus, suitable K supplementation improves the ability of the plants to cope with drought stress through the efficient use of water [112]. Proline accumulation is one of the major mechanisms of drought tolerance in plants [113,114]. Stress tolerance through the accumulation of proline has been reported to be intensified through the foliar spray of K [115,116]. Potassium deficiency has also been linked to a significant reduction in the rate of photosynthetic CO_2_ fixation, partitioning and use of photosynthates in moisture-deprived conditions [117,118].

Researchers have also documented the ameliorating effect of exogenous K application in field crops under waterlogging by enhancing plant height, photosynthetic ability, chlorophyll content, nutrient uptake and antioxidative activity while reducing lipid peroxidation [119,120]. Additional K fertiliser application of 160 kg ha^−1^ was found to mitigate the water stress effect in rice by improving yield harvest index and other physiological parameters [38]. Application of K has been found to regulate osmotic adjustment, cytoplasmic homeostasis, membrane potential and enzyme activation under salinity stress [121,122]. Application of K helps in escalating potassium-sodium ionic ratio (K^+^/Na^+^), which in turn facilitates higher affinity K^+^ transporter-mediated specific transport of Na^+^ as well as co-transport of Na^+^–K^+^, inducing Na^+^ tolerance [123]. The deficiency of K has been reported to exert several negative impacts, such as chlorotic and necrotic disorders in plants exposed to higher intensities of light [2]. On the other hand, with a lack of K fertilisers, freezing or chilling temperature-induced enzymatic dysfunction, disrupted fluid transport, carbon absorption, and photooxidative injury increase, impairing the normal growth and productivity of crop plants [124,125]. Sufficient K supply helps in eliminating freezing-induced dehydration by adjustment of osmotic potential and better ROS defence mechanisms [126].

### 2.2. Secondary Nutrients

The functional aspects of secondary nutrients, i.e., calcium (Ca), magnesium (Mg) and sulphur (S) in the activation of different plant mechanisms to ameliorate abiotic stress are depicted in Figure 3. Some prominent crop-based examples regarding the impacts of these three elements under stress situations are presented in Table 3.

#### 2.2.1. Calcium

Calcium (Ca) is an essential secondary nutrient, mediating the cell and plant development processes. It also improves plant response to different stress conditions by regulating many physiological aspects [144]. It is also a secondary messenger element for intracellular processes, as it acts as a signalling molecule in different physiological and biochemical pathways in the plant to develop stress resistance [145]. Calcium is also important for nutrient uptake, hormonal and enzymatic upregulations and stabilisation of cellular membranes to mitigate abiotic stress in plants [146]. There are reports that Ca reduces yield loss in different crops under diverse abiotic stress conditions, including salt, drought, flooding, heat, chilling, and heavy metal stress [147]. The reactive oxygen species (ROS) act as signalling molecules for moderating stress tolerance by driving the Ca^2+^-governed stress-responsive genes [148]. Cytolic Ca^2+^ increases quickly under stressful conditions and is dependent on Ca^2+^ binding proteins, such as calmodulin [149]. The Ca^2+^ binding protein, through the enhanced cystolic Ca^2+^ signal, subsequently adjusts and protects the responses in plants under adverse conditions [147]. The Ca-dependent protein kinases (CDPK) stimulate the stress-responsive genes and regulate abiotic stress physiological responses such as stomatal movement, K^+^ uptake and gene expression [150,151], thereby playing a significant role in antioxidative stress response [152].

Under salt stress, Ca ions can ameliorate its effect by competing with Na^+^ ions for the membrane binding sites [153]. The study by Tuna et al. [154] showed that the application of calcium sulphate (CaSO_4_) enhanced the concentration of Ca^2+^, N, and K^+^ and reduced the concentration of Na^+^ in the leaves when grown in pots under salt stress. The adverse effects of salt stress on seed germination can be reduced under saline conditions as Ca restricts the entry of Na^+^ ions [155,156]. The effect of salinity stress on germination was alleviated by the application of Ca in *Pisum sativum*, *T. aestivum*, *H. annuus*, *L. esculentum* [157,158] and *Chenopodium album* [159]. Patel et al. [160] confirmed that supplementing Ca in *Caesalpinia crista* L. in salinized soils restored the levels of reduced N, P, K, and Ca content in tissues. Rice is reported to be highly sensitive to salt stress. The early seedling stage is the most salinity-sensitive growth stage that directly affects the yield. In total, 10 mM CaCl_2_ supplementation on salt-stressed rice seedlings in the early vegetative stage increased the chlorophyll and proline content and oppressed the accretion of ROS, thus protecting them from oxidative damage in salt-susceptible varieties [127]. With the increase in external Ca concentration, the concentration and uptake of Na decrease and Ca concentration and uptake increase, as Ca^2+^ restricts Na^+^ uptake and it interferes with the non-selective cation channel [153].

Seed pelleting with calcium oxide (CaO) increased the percentage of seed germination as well as the seedling growth under waterlogging conditions [161] due to increased oxygen availability [147]. Calcium also plays a significant role in plant drought resistance. Ca enhances the ability to conserve water when applied to leaves [162]. Ca^2+^ changes the degree of hydration of the plasma membrane, thereby improving the cohesion of the cell walls, which thereby increases the resistance of cells to dehydration by increasing the viscosity of protoplasm [163]. Xu et al. [164] observed that Ca increased root and shoot biomass along with dry weight in *Zoysia japonica* with the application of 10 mM Ca under drought conditions. Berkowitz et al. [165] showed that during water stress, Ca^2+^ hinders the influx of K^+^ ions to the guard cells by affecting inward K^+^ channels [153]. Calcium regulates plasma membrane ATPase, which is required to pump nutrients back that were lost in the case of cell damage and thereby plays a significant role in recovery from drought-induced damage [166]. Under the dark condition, the inhibiting effect on germination of *P*. *karka* [155], and *U. setulose* [167] was reduced by the addition of Ca^2+^. Nayyar and Kaushal [168] demonstrated that the chilling-induced oxidative stress in the seeds of wheat (*T. aestivum*) can be partially mitigated by Ca [147] by reducing lipid peroxidation and membrane damage.

#### 2.2.2. Magnesium

Magnesium (Mg) serves as a structural component of the ribosome and is fundamental for the conformational stabilisation of macromolecules such as nucleic acids, proteins, cell membranes and walls [169,170]. As Mg is an essential component in the chloroplast that regulates photosynthetic activity, its deficiency can affect photosynthesis. It is essential for the maintenance of enzyme activities such as ATPase, kinases and polymerases [171]. Magnesium plays a significant role in the activity of every phosphorylating enzyme in carbohydrate metabolism [172,173]. This nutrient is also a cofactor of enzymes involved in metabolism and photosynthetic carbon fixation [174]. The cation-anion balance in the cells is regulated by Mg, which, along with K, acts as an osmotically active ion-regulating cell turgor [175,176].

The bioavailability of Mg is reported to be decreased by heat [134] and drought stress [177]. It is observed in general that plants when exposed to salt stress exhibit decreased levels of Ca^2+^, Mg^2+^ and K^+^ [178]. Ferreira et al. [179] reported that in the stems and roots of *Psidium guajava* L., salinity did not affect the Mg^2+^ content, but it decreased in the leaves. Barhoumi et al. [180] reported that in *Aeluropus littoralis*, salinity stress did not affect Mg content. A high level of Mg in leaves can help in maintaining better water content during drought. Under drought stress, *Musa acuminate* plants showed an accumulation of around 28% higher Mg content than the control [181].

#### 2.2.3. Sulphur

Sulphur (S) is the fourth most important plant nutrient after N, P and K, and is an essential macronutrient in plants that serves various functions. Many S-containing chemicals play protective functions in abiotic stress response, cellular acclimatization, and adaptability in adverse conditions [182]. An exogenous supply of S has been shown to benefit plants’ survival in stressful conditions by maintaining their normal metabolic processes and also improving crop yield [107]. Sulphur is integrated into cysteine (Cys) after being taken up by the roots in the form of sulfate (SO_4_^2−^). Cysteine acts as a precursor or donor of key S compounds such as methionine (Met), S-adenosylmethionine, glutathione (GSH), homo-GSH (h-GSH), phytochelatins (PCs), sulfolipids, iron-sulfur clusters, allyl Cys, and glucosinolates, which play a role in plant developmental processes and/or stress adaptation processes [183,184]. GSH, hydrogen sulfide (H_2_S), Met, Cys, PC, ATP-S, protein thiols, and other sulfur-containing compounds play a significant role in the normal functioning of the plant cell. Stress signal transmission is aided by GSH, which is one of the most effective antioxidants and stress protectors [185,186].

Min et al. [187] reported that S helps to mitigate heat stress by increasing activities of SOD, CAT and APX; higher H_2_S and soluble sugar contents but reducing H_2_O_2_ and MDA contents. In *Arabidopsis thaliana*, the application of 1500 μM S as SO_4_^2−^ helped alleviate the adverse effect of soil salinity by upregulating some antioxidant enzymes and maintaining ABA level [182]. Sulfate nutrition regulates arsenic (As) translocation from roots to shoots, potentially through the complexation of As III-PCs [188]. Sulphur also protects against Mn toxicity by increasing antioxidant defence and improving Mn transport and distribution from roots to shoots [189]. Stress signalling is enabled via interactions of S with other biological molecules, which provides resistance against external pressures. However, S absorption, transport, and mechanisms of action in stressed plants are still being investigated and need to be validated.

### 2.3. Micronutrients

A jest mechanism by which the micronutrients, namely, boron, zinc, iron and copper, mitigate abiotic stress is given in Figure 4. While some specific crop-based instances regarding the effects of these four elements under stress situations are given in Table 4.

#### 2.3.1. Boron

Boron (B) application has a significant impact on reducing the negative effects of abiotic stress, improving yield, and nutrient uptake. Boron plays a vital role in the overall metabolism and transport system of carbohydrates as well as synthesis and functional aspects of cellular integuments [116,210]. Application on B mitigates the negative effects of saline environments by maintaining internal K^+^ balance through retaining cell wall elasticity and recovery of proper levels of K^+^ [211]. Moreover, B supplementation is a well-established remedy to recover nutrient balance, while improving salt tolerance of nitrogen-fixing leguminous plants [212]. It also promotes the resistance of crop plants against drought stress by improving photosynthetic efficiency, hormone synthesis, sugar transport, lipid metabolism, flower retention, pollen formation, seed germination and seed yield [191].

Plants with sufficient B nutrition have shown an elevated resistance to drought stress due to improved nutritional status and enhanced water uptake from the rhizospheric soil by growing more root hairs and mycorrhizae. Boron can affect the drought sensitivity of crop plants in two different ways. Firstly, it is involved in the ROS detoxification process in chloroplasts and thereby takes a protective role in the prevention of photooxidative damage catalysed by ROS. Secondly, B may contribute to drought tolerance by protecting against oxidative damage to cell membranes [212]. In particular, the tolerance mechanism to moisture deficit is attributed to a rise in total glutathione and ascorbate pools, which control the accumulation of hydrogen peroxide and prevention of electrolyte leakage in the plasma membrane along with the impairment of gaseous exchange.

The role of B in cell wall structure formation, sugar translocation, membrane integrity and plant reproductive growth is critical for reducing the damage caused by abiotic stress, particularly high-temperature stress [118]. In a study by Shahid et al. [211], it was found that application of 1 and 2 kg B ha^−1^ at vegetative and reproductive stages reduced the negative impacts of a high temperature up to 37.6 °C. They reported that exogenous application of B stabilised the cellular membranes, mobilised carbohydrates and improved pollen grain, thus alleviating temperature stress.

Alike high-temperature stress, chilling stress also leads to oxidative stress that increases ROS, negatively affects membrane lipids and ultimately causes plant cell death. Chilling stress reduces leaf expansion and growth, wilting and chlorosis, which may lead to a lower photosynthetic surface. Boron application mitigates chilling stress [213]. During chilling stress, application of B can enhance photosynthetic activity and enhance the activities of plant antioxidants. This helps to reduce ROS injury caused by temperature. Boron nutrition also improves sugar transport in the plant body, which acts as an osmolyte and anti-freezing agent [214].

Foliar application of B was found more effective to alleviate the deleterious effect of waterlogging [215]. The application of boron increased the stability of leaf membranes, chlorophyll, soluble sugars, soluble proteins, amino acid contents, LRWC and dry mass accumulation [216].

#### 2.3.2. Zinc

Zinc (Zn) nutrition facilitates better defence against heat stress by maintaining the membrane integrity inside the plant system [217]. Zn is an integral constituent of plant enzymes such as carbonic anhydrase, alkaline phosphatase, alcohol dehydrogenase (ADH), RNA polymerase, Cu-Zn SOD and phospholipase. It has been identified as a major component of different proteins associated with DNA and RNA synthesis [217]. Peck and McDonald [13] observed that lower Zn supply significantly hampered the plant growth process, while an adequate supply of Zn minimised the negative effects of heat stress. Antioxidant properties such as SOD activity in the wheat plant were found to be more stable in the presence of Zn under heat stress. The plants’ deficit of Zn showed higher sensitivity towards heat stress as compared to adequate Zn application. Basically, zinc application resulted in enhanced activity of SOD, APX, glutathione reductase and glutathione peroxidase in the Varuna cultivar of mustard and resulted in 145% more yield [183].

Zinc has also proved its role in alleviating chilling stress [218,219]. Zinc plays a key role in upholding the structural integrity of proteins, membrane lipids, various cell components and DNA, along with facilitating ion transport in plants [220]. Foliar application of zinc oxide (ZnO) nanoparticles (NPs) can efficiently mitigate the toxicity of chilling stress. The toxicity of the chilling effect in rice was mitigated through the regulation of the genetic expression of the transcription factors related to chilling stress response by foliar application of ZnO [221]. Evidences are also found regarding the influential and multiple effects of ZnO NPs on plant growth and chlorophyll biosynthesis, eventually enhancing the antioxidant potential and ROS scavenging abilities under chilling stress [222].

Harris et al. [223] reported improved germination and yield of maize, wheat and chickpea under a wide range of environmental conditions through seed priming with Zn. Drought stress reduces plumule length and shoot dry weight owing to restricted remobilisation of nutrients from photosynthates reserves to the embryo. The intervention of Zn priming has been observed to improve the synthesis of plant hormones including IAA and gibberellic acid (GA_3_) under moisture stress conditions and thereby augmenting plumule characteristics under drought stress [174]. Adequate Zn supply under drought stress regulates membrane permeability, activity of antioxidant substances and enhances photosynthetic efficiency and water use efficiency. In addition, Zn application helps in a significant expansion in leaf area, improvement in photosynthetic pigments such as chlorophyll and others, stomatal conductance, relative leaf water content and osmolyte accumulation, thus resulting in improved growth, yield and prevention of leaf tissues from the destructive impacts of moisture deficiencies [224].

Zn application improves plant tolerance to salt stress by stabilising root-cell membranes while preventing ion leakage from roots by limiting root permeability [225,226]. For instance, the application of Zn resulted in better yield in salinity-stressed chickpea through management of osmotic stress, ionic balance and prevention of solute leakage [227]. In maize (corn), foliar application of 1% ZnSO_4_ at tassel initiation and grain filling increased the thousand kernels’ weight from 27.3 to 31.3 g and induced an increase in the number of seeds per year from 710 to 770, apart from providing resistance against drought stress [196].

During waterlogging conditions, plants suffer from severe hypoxia and subsequent inhibition in ATP formation because the rate of O_2_ diffusion is much slower in water compared to that in air. Thus, flooded plants are required to shift their carbohydrate metabolism towards fermentation and up-regulation of genes for ADH and pyruvate decarboxylase to sustain cellular energy levels at optimum [228]. Application of Zn fertilisers improves plant tolerance toward waterlogged conditions under Zn deficiency by reenergizing the diminished ADH activity [229].

#### 2.3.3. Iron

Iron (Fe) is one of the chief components of the cell redox systems and also functions as a cofactor regarding various antioxidant enzymes such as CAT, POD and APX [230,231]. In plants, Fe assists in chlorophyll synthesis and is essential for the maintenance of structural integrity as well as functional aspects of the chloroplast. The Fe-S cluster serves as a prosthetic group for several Fe-S proteins and plays regulatory roles during oxidative stress situations [232]. Therefore, Fe application has a crucial role in the life cycle of plants under stressed conditions.

The application of Fe as a plant nutrient exhibited a significant ameliorative effect against salt stress by producing antioxidative enzymes. These antioxidative enzymes include CAT, POD and SOD that act as major scavengers of ROS, thereby strengthening cell defence mechanisms against salinity [233]. Ghasemia et al. [234] suggested the protective role of Fe^2+^ in tomato plants against salinity through the formation of chelates with amino acids.

The provision of Fe nutrition to plants under drought conditions can increase stress tolerance as it leads to assimilate synthesis [235,236,237]. In this context, legumes have also been reported to develop positive responses to Fe nutrition [238]. Baghizadeh and Shahbazi [239] reported that foliar Fe nutrition with Zn reduces oxidative stress by depleting H_2_O_2_ content along with bringing down lipid peroxidation by accelerating antioxidant enzyme mechanisms (CAT, GPX and SOD) under drought situations. Spraying with Fe also plays a significant role in improving the quality and resistance of protein under drought stress [240]. Foliar spraying of B @ 0.2% + Fe @ 0.5% produced 58.3% and 27.0% higher seed and stover yields than the control treatment apart from alleviating combined heat and moisture stress [239].

Iron oxide nanoparticles have the ability to alleviate the negative effects of cold stress by lowering electrolyte leakage and membrane damage. Iron oxide nanoparticles increase chlorophyll and Rubisco-binding protein genes, thus increasing cold stress tolerance [241]. Atar et al. [242] also pointed out the role of iron oxide nanoparticles in mitigating cold stress in plants by stimulating certain enzymes such as catalase, cytochrome oxidase and peroxidase. The application of iron nanoparticles as a chilling treatment improves the biochemical, physiological, and growth attributes of plants under cold-stressed conditions.

Iron toxicity is a principal constraint, commonly encountered under waterlogged conditions. Under anaerobic environments, reduced Fe is massively absorbed by plant roots, inducing the generation of ROS. This oxidative stress is responsible for substantial growth reductions, physiological perturbations and drastic yield losses [243]. The hampered efficiency of PS II is attributed to the deficiencies of N, P, K, Mg and Ca owing to deficit Fe [244].

#### 2.3.4. Copper

Copper (Cu) is essential for plant growth and the activation of a number of enzymes. The deficiency of Cu interferes with protein synthesis and leads to a subsequent build-up of soluble nitrogenous compounds. Copper is a redox-active transition element that plays a significant role in photosynthesis, C and N metabolism, respiration and protection against oxidative stress [245]. Plastocyanin, which is related to the photosynthetic electron transport pathway in PS I and occurs in the thylakoid lumen of leaf chloroplasts, includes the most Cu [246]. Cu/Zn SOD, a significant Cu protein, contributes to the ROS scavenging mechanisms [247]. Copper chlorophyllin (Cu-chl), an important modified bio-stimulant which is a water-soluble and semi-synthetic derivative of chlorophyll, has been found to reduce oxidative stress by means of its presumed strong antioxidative capacity [248]. Cu-chl has been shown to be protective against the effects of drought stress in tomatoes, where the frequent application of Cu-chl-containing materials improved the activity of leaf antioxidant enzymes as well as GSH concentration [249]. Islam et al. [250] reported other functions of Cu-chl with respect to the development of tolerance against high salinity stress in *A. thaliana*. Soil salinity significantly decreases RWC with a concomitant increase in lipid peroxidation, total phenolics and cell membrane permeability. The application of Cu successfully mitigates the adversities of salinity stress through the protection of cellular membrane damage and reduction of lipid peroxidation while increasing RWC [251]. The colloidal suspension of Zn-Cu nanoparticles was found to have a beneficial impact under moisture deficit conditions. Copper nanoparticles applied under drought declined reactive substances and increased antioxidative enzyme activity [252]. The colloidal suspension of Zn-Cu nanoparticles increases the chlorophyll content and carotenoid content in the leaves, which is an adaptive mechanism of the plant to drought conditions [253].

### 2.4. Beneficial Nutrients

The functional aspects of beneficial nutrients, i.e., cobalt (Co), selenium (Se) and silicon (Si) in the regulation of plant physiological processes to relieve various abiotic stresses are illustrated in Figure 5. Moreover, the detailed effects of these three beneficial nutrients on different crops under stress situations are delineated in Table 5.

#### 2.4.1. Cobalt

Cobalt (Co) exemplifies an impressive example of a beneficial element to regulate vital physiological and metabolic functions in plants, with special reference to leguminous crops [264,265]. It synthesises leghaemoglobin protein for the purpose of rhizobial activity as well as biological nitrogen fixation in legumes, thereby exerting a significant impact on enzyme systems [266,267]. Cobalt also imparts drought tolerance in plants by virtue of efficient utilisation of water in addition to reducing the rate of transpiration [268,269]. The substantial effect of Co has been reported to curb moisture stress by activating the antioxidant defence mechanisms in plants [270,271] under heat and drought-stressed conditions [116]. In fact, it stimulates the activities of amino acids and enzymatic antioxidants such as SOD, CAT, POD and polyphenol oxidase (PPO) while delaying leaf senescence by obstructing ethylene biosynthesis [272,273]. A number of literatures have documented the ameliorative role of Co on the adversities of heat, moisture and heavy metal stresses on field crops through leaf area expansion, chlorophyll synthesis, osmotic adjustments, maintaining membrane integrity, efficient use of thermal energy and activation of antioxidant defence mechanisms [113,274]. Alongside this, the adversities related to soil salinity stress in plants can be moderated by means of Co application by promoting normal physiological, biochemical and metabolic activities [266,275]. Some authors have also reported the ameliorative effects of Co in relation to cadmium toxicity-induced retardation of photosynthesis and transpiration rates by suppressing oxidative stress [276,277].

#### 2.4.2. Selenium

Selenium (Se) has been recognised as an essential trace element at a relatively lower concentration with multifarious beneficial impacts on plants [278,279]. It has been established as a strong phytoprotectant when the plant is exposed to different environmental adversities [280,281,282]. Application of Se can induce tolerance to abiotic stresses alone or in combination with several other plant nutrients [283] by retarding senescence and regulating water economy [284], photosynthesis [285] and Na^+^ homeostasis [286], thereby promoting growth [287]. Del Pino et al. [288] documented the persistent impact of Se fertilisation on the augmentation of qualitative aspects of crop plants under abiotic stress, especially under Se-deficient zones. Specifically, it rejuvenates ROS scavenging mechanisms through activation of antioxidant defence [289] such as GPX, APX, DHAR, MDHAR, CAT, POD and SOD activities, while curtailing the H_2_O_2_ and MDA contents under different kinds of abiotic stresses [290,291]. Conjoint activities of these antioxidants reduce lipid peroxidation as well as retain cell membrane integrity and photosynthetic pigment concentrations [292,293]. Selenium is reported to protect crops against drought and heat stress [294], chilling stress [295], salt stress [296], acid stress [297], UV ray-induced oxidative stress [298] etc. Interestingly, nanoparticles of Se have been registered to improve the growth attributes, photosynthetic ability and antioxidant defence of crops for scavenging ROS under heat and chilling stress, with special reference to sorghum [299] and tomato [300]. There are also some other reports where the significant role of Se has been elucidated through modulations in metabolites transport, photosynthetic activity, membrane integrity and cellular turgor in ameliorating drought, salinity and heavy metal stress in the cases of rice [301], wheat [302], maize [258], onion [303], carrot [304], cucumber [305] etc. However, the beneficial effects of Se in terms of plant growth have been attributed to the antioxidative property of the element itself [285,306] by virtue of higher accumulation of soluble sugars and proline in the shoots [307,308] along with higher LRWC, biosynthesis of photosynthetic pigments and enzymatic upregulations to strengthen the metabolic balance inside moisture-stressed plants [309].

#### 2.4.3. Silicon

Silicon (Si) is another important beneficial trace element for plant growth as well as abiotic stress alleviation [310,311,312]. The application of Si nutrition boosts the antioxidant enzyme production, including SOD, CAT and POD, while reducing ROS generation [313]. In fact, Si alleviates oxidative stresses by the accumulation of protective proteins as well as decreasing the metal-ion toxicity, thereby amplifying the efficiencies of oxidative enzymes, while eliminating O_2_^−^ and H_2_O_2_ from the cells and reducing lipid peroxidation [314,315]. Abdel Latef and Tran [262] reported that seed priming with a silicon-enhanced LRWC and levels of photosynthetic pigments, soluble sugars, soluble proteins, total free amino acids and K^+^, as well as activities of SOD, CAT, and POD enzymes, finally resulted in a higher yield in the stressed crop. Together, these factors reduce the severity of photooxidative damage and protect the integrity of cellular membranes, which enhances plants’ drought tolerance mechanisms [314,316]. Silicon plays a significant detoxifying role under aluminium stress [317] and cadmium stress [318] by stimulating the biosynthesis of proline and phenolic compounds. Although the mechanistic understandings of the functional role of Si in abiotic stress tolerances in plants are relatively limited [319,320], various scientific reports are emerging where it has been reported that Si application encourages growth, photosynthesis, economic yield and also withstands soil salinity and drought [321,322]. Studies have claimed that the primary impact of Si is more distinct on stressed plants rather than on plants growing under normal conditions [323,324]. Several long-term experiments have executed the positive effects of Si in terms of lowering the rate of transpiration while accelerating photosynthetic activity and photoassimilateing mobilisation by modulating mesophyll conductance and thereby improving crop yield under moisture stress [325,326,327]. Moreover, the presence of Si has also been established to maintain LRWC [328], root hydraulic conductance [329], production of photosynthetic pigments [330] and levels of different osmoprotectants [331] when exposed to stress.

## 3. Combined Effects of Plant Nutrients

Despite the fact that all plant nutrients contribute in some way to plant growth and the reduction of stress, we must pay close attention to the proper nutritional composition. For optimum nutrient utilisation efficiency, antagonistic (negative) nutrient interactions should be minimised and synergistic (positive) nutrient interactions should be increased. Understanding the potential for both positive and negative interactions between nutrients is necessary for these actions. There are reports where it has been mentioned that some nutrients together can combat stress effectively, whereas some may cause negative impacts. Some of the nutrient interactions are discussed in the below Table 6 as follows:

## 4. Conclusions and Future Needs

One of the biggest challenges in agricultural production is to secure future food security. However, environmental stresses are a significant hurdle in this endeavour. Abiotic stresses affect the morphoanatomical and physiological growth of plants. In general, these stresses affect chlorophyll synthesis, leaf growth, enzyme activity, transpiration, stomatal conductance, membrane stability and finally crop productivity. A changing climate further increases the detrimental effect of abiotic stress on plants. Though developing variety is an essential step for adapting, using the correct plant nutrients can help plants develop abiotic stress tolerance. We have tried to explain the various mechanisms by which plant nutrition alleviates various abiotic stress in this review. In general, nutrients such as nitrogen, potassium, calcium and magnesium increase the concentration of antioxidant enzymes such as superoxide dismutase (SOD), peroxidase (POD) and catalase (CAT) reducing reactive oxygen species (ROS). Nutrients such as potassium and calcium help in improving stomatal regulation and osmotic adjustments by improving water uptake. Under temperature stress, these nutrients aid in maintaining a high tissue water potential. Micronutrients such as iron, boron and zinc help in activating various physiological changes in plants, activate defence mechanisms and improve the metabolic process by which the plants adapt to various adverse stresses. There are some combinations of nutrients which together can more effectively ameliorate the stress and vice versa. Though plant nutrients are a low-cost and sustainable way of managing abiotic stresses, there is still much that needs to be further explored. A detailed research needs to be performed on the role of these nutrients under stress. In the wake of climate change, a better understanding of nutrient interaction, their optimum concentration and phenological stage of application will help for better management of abiotic stresses.

## Figures and Tables

**Figure 1 ijms-23-08519-f001:**
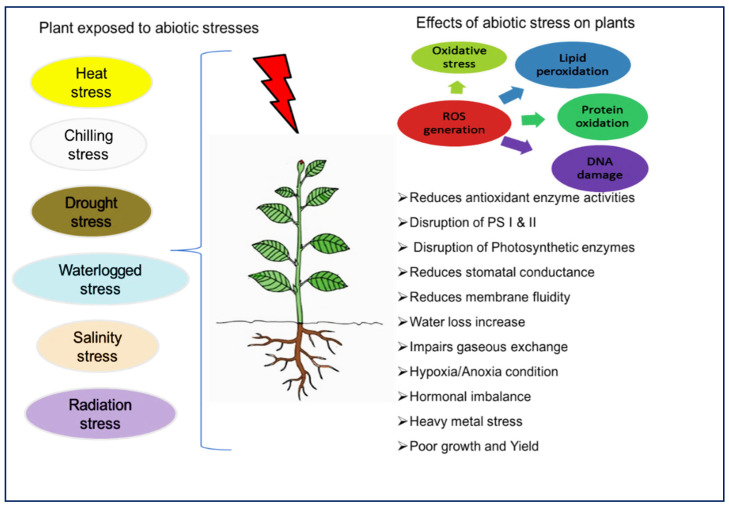
Different abiotic stresses and their effects on plants.

**Figure 2 ijms-23-08519-f002:**
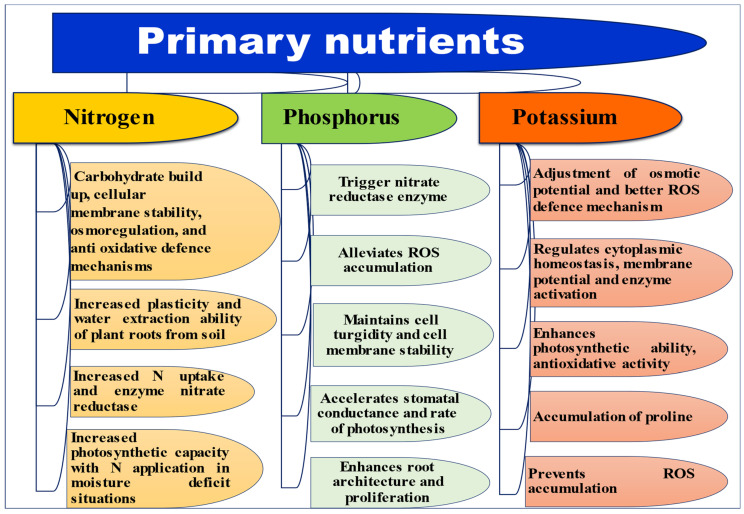
Activation of various plant mechanisms by application of primary nutrients to alleviate plant stress.

**Figure 3 ijms-23-08519-f003:**
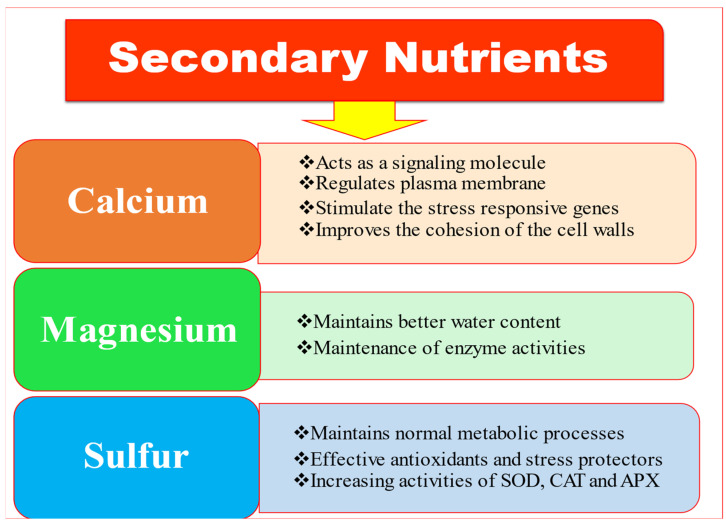
Activation of various plant mechanisms by application of secondary nutrients to alleviate plant stress.

**Figure 4 ijms-23-08519-f004:**
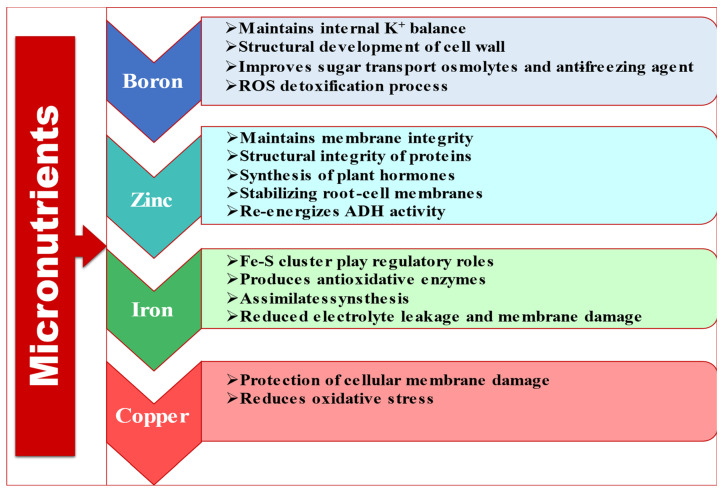
Activation of various plant mechanisms by application of micronutrients to alleviate plant stress.

**Figure 5 ijms-23-08519-f005:**
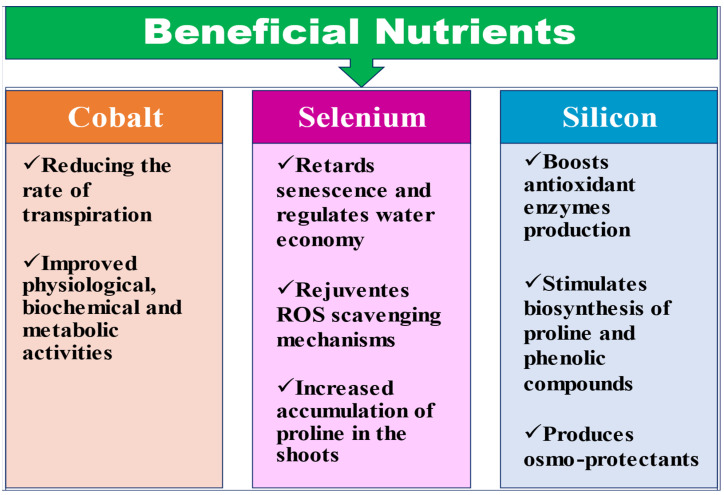
Activation of various plant mechanisms by application of beneficial nutrients to alleviate plant stress.

**Table 1 ijms-23-08519-t001:** Abiotic stresses and their effects on plant growth and activity.

Abiotic Stress	Effects on Plant	References
Heat	Causes water scarcity, osmotic and oxidative stress that enhances ROS production, protein misfolding and denaturation. Foliar senescence and leaf discoloration, reduced CO_2_ fixation and disturbed ion transport.	[14,15,16]
Chilling	Leads to osmotic and oxidative stress, nutritional imbalance. Accumulation of ROS, inhibition of enzyme activities, and reduced photosynthetic activity.	[17,18]
Salinity	Responsible for water scarcity and ionic imbalance. Osmotic and oxidative stress that enhanced ROS production, restricted uptake and translocation of water and mineral nutrients, decreased stomata opening and reduced photosynthesis.	[19,20]
Drought (Water deficit)	Causes osmotic and oxidative stress. Increased ROS production and ion leakage. Decrease in absorption and translocation of mineral nutrients. Protein denaturation, loss of enzyme activities	[21]
Flooding/water-logging	Leads to oxidative stress and increased ROS production. Reduced gaseous exchange and photosynthetic activity due to lower chlorophyll content.	[22,23]
Light/radiation	Oxidative stress, increased ROS production and oxidative damage, reduced photosynthetic activity, and chlorophyll degradation.	[24]

**Table 2 ijms-23-08519-t002:** Crop-wise effects of macronutrients applied under stress situations.

Macronutrient	Tested Crop	Mechanisms Related to Stress Alleviation	References
N	Rice	Strengthening root system, improved xylem transport,	[30]
	Wheat	upregulated photosynthesis, chloroplast fluorescence, enzymatic functions, relative leaf water content, nutrients uptake, lower lipid peroxidation, improved antioxidant defence mechanism in terms of SOD and POX	[31]
	Maize	Higher leaf expansion, reduced Na uptake, delayed cell senescence, stomatal regulation	[32]
	Rapeseed	Improved plant water status, greater proline accumulation	[33]
	Forage pearl millet	Escalated water use efficiency, profuse branching	[34]
P	Wheat	Better root and shoot extension, carbohydrate transport, increased nutrients and water use efficiencies	[35]
	Cotton	Improved relative leaf water status	[36]
	Moth bean	Enhanced nutrients uptake, root hydraulic conductance, modified root development, leaf moisture content	[37]
K	Rice	Better shoot development, greater synthesis of osmolytes	[38]
	Wheat	Improved antioxidant enzymatic functions	[39]
	Oat	Augmented nitrogen metabolism and antioxidant defence system	[40]
	Indian mustard	Higher leaf area expansion, membrane stability, leaf water status, modified antioxidants activity regarding CAT, POX, APX, SOD	[41]
	Cotton	Enhanced photosynthetic mechanism, improved carbohydrate metabolism	[42]

**Table 3 ijms-23-08519-t003:** Crop-wise effects of secondary nutrients applied under stress situations.

Secondary Nutrient	Tested Crop	Mechanisms Related to Stress Alleviation	References
Ca	Rice	Improved germination characters, shoot and root development, enhanced leaf chlorophyll and proline contents, oppression of ROS by stimulation of CAT and POX	[127]
	Maize	Water regulation at cellular level and root development	[128]
		Improved growth, osmotic relation and proline content, reduced H_2_O_2_ activity	[129]
	Barley	Alleviation of Al toxicity, ROS suppression and antioxidative enzymes	[130]
	Sugarbeet	Extensive leaf coverage, chlorophyll content, carbohydrate accumulation, reduced oxidative stress by escalating glutathione and free polyamine putrescine pools while reducing amino acid gamma-aminobutyric acid levels	[131]
	Tobacco	Accelerated photosynthetic activity, stomatal conductance, improved thermostability of different oxygen-evolving complex and less accumulation of ROS	[132]
Mg	Rice	Efficient sugar partitioning, better root proliferation	[133]
	Wheat,	Restricted ROS production, reduced peroxidative damage in leaf chloroplasts, improved antioxidative defence enzymes	[134]
	Maize		
	Broad bean	Increased activity of plasma membrane ion transporters, improved light-induced responses of leaf mesophyll	[135]
	Mungbean	Greater synthesis of photosynthetic pigments, higher proline accumulation	[136]
S	Rice	Alleviation of As toxicity through improved amino acids, proteome and thiol metabolisms	[137]
		Alleviation of Cd toxicity by means of increasing Fe plaque formation, Cd chelation and vacuolar sequestration	[138]
	Barley	Reduction of salt accumulation, regulation of NO signalling and ion homeostasis	[139]
		Alleviation of Al toxicity through stimulation of ATPase activity and reducing oxidative stress	[140]
	Oilseed rape	Improved photosynthesis, nutrient uptake	[141]
	Mustard	Enhancing leaf ascorbate and glutathione levels, amelioration of Cd toxicity,	[142]
		Improved photosynthesis, salt tolerance through greater glutathione production	[143]

**Table 4 ijms-23-08519-t004:** Crop-wise effects of micronutrients applied under stress situations.

Micronutrients	Tested Crop	Mechanism to Mitigate Abiotic Stress	References
B	Chickpea	Increased antioxidative enzymes, such as SOD, CAT and APX	[190]
	Cowpea	Increase in SOD activity, photosynthesis, leaf chlorophyll content	[191]
	Sunflower	Increase in SOD activity, increases photosynthetic activity	[192]
	Tomato	Increase in SOD activity	[192]
	Rice	Increase in the enzymatic activity of APX, POD and CAT.	[193]
	Potato	Increase in leaf proline, protein, carbohydrates and antioxidant enzymes such as polyphenol oxidase and peroxidase in tubers	[194]
Zn	Maize	Higher plant biomass, stomatal conductance and quantum yield of photosystem II. Improved grain yield, RWC and chlorophyll content under drought stress.	[195,196]
	Wheat	Higher chlorophyll content and activities of SOD, POD and CAT at grain filling stage, quantum yield of PS-II, chlorophyll content, stomatal conductance,	[197,198]
	Chickpea	CO_2_ assimilation rate, proline content and activities of SOD, APX. Increases chlorophyll and carotenoid contents, seedling vigor and seed yield. Reduces MDA contents	[199,200]
	Common bean	Higher shoot biomass, chlorophyll and carotenoid contents, leaf NPK content Reduces MDA contents	[201]
	Tomato	Improved stomatal aperture and chlorophyll content	[202]
	Sunflower	Higher chlorophyll, proline contents and SOD activities	[203]
Fe	Rice	Higher water content, higher activity of hydrolytic enzymes amylase and protease, increased activity of SOD, CAT and glutathione peroxidase, increased cell membrane integrity, cell viability, chlorophyll and iron content, increased activity of NADPH dehydrogenase	[204]
	Wheat	Increased root length, biomass growth and chlorophyll content, increased activity of peroxidases and SOD, decreased level of MDA	[205]
	Sunflower	Increased activity of antioxidant enzymes, superoxide dismutase, peroxidase, catalase and ascorbate peroxidase	[206]
	Soybean	Enhanced net photosynthetic rate, stomatal conductance, intercellular CO_2_ concentration, transpiration rate, increased shoot weight	[207]
Cu	Tomato	Increased antioxidant activity, phenols, vitamin C, glutathione, and improved Na+/K+ ratio	[208]
	Maize	Increased photosynthesis, water relation, osmotic adjustment, decreased membrane damage and lipid peroxidation, increase in RWC	[209]

**Table 5 ijms-23-08519-t005:** Crop-wise effect of different beneficial nutrients applied under stress situations.

Beneficial Nutrients	Tested Crop	Mechanisms Related to Stress Alleviation	References
Co	Wheat	Increased chlorophyll content, chlorophyll stability index and proline content	[254]
	Maize	Enhanced physiological efficiency	[255]
	Black gram	Improved chlorophyll and carotenoid contents, proline and nitrate reductase contents, better cell membrane stability	[116]
	Soybean	Enhanced antioxidant activities	[256]
Se	Maize	Accelerated net photosynthetic rate, greater integrity in chloroplast ultrastructure	[257]
	Rapeseed	Rejuvenation of entire antioxidant system in terms of APX, MDHAR, DHAR, GR, GST, GPX, CAT, and glyoxalase I and II, higher ROS scavenging, reduced membrane peroxidation and subsequent lower production of MDA	[258]
Si	Rice	Increased nutrients uptake, improved plant growth, yield, and quality	[259]
	Wheat	Better regulation of antioxidant enzymes	[260]
	Maize	increased water use efficiency, reduced leaf transpiration	[261]
		Enhanced leaf relative water content, concentrations of photosynthetic pigments, soluble sugars, soluble proteins, free amino acids, improved K/Na ratio by reducing, Na uptake	[262]
	Sorghum	Higher water uptake, greater rate of assimilates transport	[263]

**Table 6 ijms-23-08519-t006:** Nutrient interactions under abiotic stress.

Interacting Nutrients	Synergistic Effect	Antagonistic Effect	References
N, P, K, B, Co	Increased growth rates, accelerated enzymatic activity, proline content and cell membrane stability	-	[116]
N, P, K, Mo	Improved growth, physiological efficiency, nutrients uptake and yield ameliorate heat and moisture stress	-	[332]
K and Ca	Improved physiological, biochemical and molecular mechanisms ameliorating drought, salinity and cold stress	-	[333]
Zn, B and Si	Increased plant height, shoot dry weight, number of stems per plant, leaf relative water content, leaf photosynthetic rate, leaf stomatal conductance, chlorophyll content, and tuber yield		[194]
B, Fe, Zn	Improved chlorophyll biosynthesis, photosynthetic rate, gaseous exchange regulation and osmoregulation to mitigate the abiotic stress in late sown lentil.Scavenge ROS, enhance antioxidant enzyme activity in chloroplast, maintain membrane integrity and decrease lipid peroxidation	-	[334]
Fe and Mn	Improved photosynthetic activity in plants ameliorating heat stress	-	[335]
Co with N/P/K/S/Zn/B/Mo/Ni/Sn	Greater uptake and utilisation of reserved and applied Co alleviating salinity, heat and moisture	-	[336]
Co with Ca/Mn/Fe/Cu/Cr/Cd	-	Immobilisation of available soil Co and thereby reduction in uptake of Co preventing heavy metal stress	[271]
B, Se	The combined application was more effective enhancing the activity of MDHAR and GR under salt stress, combined spray enhanced the enzymatic activities (APX, MDHAR, DHAR, GR, CAT, GPX, GST, POD) under salt stress		[291]

## Data Availability

Not applicable.

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
