# Peer review of "Plant Nutrition: An Effective Way to Alleviate Abiotic Stress in Agricultural Crops"

_ijms, 2022, doi:10.3390/ijms23158519_

Round 1
Reviewer 1 Report
In this manuscript, the authors review and discuss the effective crop nutrition techniques for alleviating abiotic stress. In this review article, the authors organize and write about the possible regulation methods of the nutrients required for the plants under stress conditions. The aims of this manuscript are clear. I don’t have major comments. However, I think the structure of this manuscript can be improved before being published. The authors described each element in this section, but the combined effect does not show in this review article. If the authors can address this point, I suggest this manuscript can go to the next step to publication.
Reviewer 2 Report
The manuscript is attempting to review the interactions between plant nutrition and abiotic stress tolerance. Some interesting info are reviewed, yet I have several major issues to raise.
The overall draft it is not well written. It is full of typos and convoluted sentences. English should be reviewed and greatly improved (just the abstract has several sentences that should be improved, a full stop is missing. Could you explain the concept trying to associate growing population and food productivity? What is food productivity?)
Most of the information in the review are not interpreted and discussed adequately with only a list of results from previous studies reported. What is lacking are further hints and insights on what can be achieved in terms of productivity by properly weighing crop fertilization.
Abiotic stresses (you claimed three times in the initial part of the draft that “abiotic stresses for instance, drought, temperature aberrations, salinity, alkalinity and heavy metal stresses can have overwhelming impacts on growth and productivity”, I agree, but what is a temperature aberration? What is alkalinity per se? you mean soil alkalinity?) impact yield but you do not present or review the effects of fertilization on plant productivity (% compared to control, which yield component can be improved, at which phenological stage can application of specific nutrient may be beneficial for maintaining or improving yield and the underlying mechanism).
Abiotic stresses in this review are oversimplified. Drought, salinity, waterlogging and temperatures are dynamic, can impose over different phenological stages when specific yield components and vegetative growth are in competition for assimilates. Considering the intraspecific complexity, the complication in defining an abiotic stress in the agricultural system (i.e. is chickpea temperature threshold the same when compared to a more cool adapted crop such as bread wheat or canola?) and the several GxE a profound rethinking regarding the concepts expressed in this manuscript should be carried out
Round 2
Reviewer 2 Report
The manuscript has been improved.
Author Response
Reviewer's comment: The manuscript has been improved
Authors' response: The authors are pleased to know that